# Effects of integrated hospital treatment on the default mode, salience, and frontal-parietal networks in anorexia nervosa: A longitudinal resting-state functional magnetic resonance imaging study

**Motoharu Gondo**[1]*, **Keisuke Kawai**[1,2], **Yoshiya Moriguchi**[3], **Akio Hiwatashi**[4], **Shu Takakura**[1], **Kazufumi Yoshihara**[1,5], **Chihiro Morita**[1], **Makoto Yamashita**[1], **Sanami Eto**[5], **Nobuyuki Sudo**[1,5]

1 Department of Psychosomatic Medicine, Kyushu University Hospital, Fukuoka, Japan, 2 Department of Psychosomatic Medicine, Kohnodai Hospital, National Center for Global Health and Medicine, Chiba, Japan, 3 Department of Behavioral Medicine, National Institute of Mental Health, National Center of Neurology and Psychiatry, Tokyo, Japan, 4 Department of Clinical Radiology, Graduate School of Medical Sciences, Kyushu University, Fukuoka, Japan, 5 Department of Psychosomatic Medicine, Graduate School of Medical Sciences, Kyushu University, Fukuoka, Japan

* gondo.motoharu.015@m.kyushu-u.ac.jp

**Data Availability Statement:** Due to the limitations of the consent provided by the subjects in our study, we cannot disclose the data to the public.

## Abstract

The psychopathology of patients with anorexia nervosa has been hypothesized to involve inappropriate self-referential processing, disturbed interoceptive awareness, and excessive cognitive control, including distorted self-concern, disregard of their own starvation state, and extreme weight-control behavior. We hypothesized that the resting-state brain networks, including the default mode, salience and frontal-parietal networks, might be altered in such patients, and that treatment might normalize neural functional connectivity, with improvement of inappropriate self-cognition. We measured resting-state functional magnetic resonance images from 18 patients with anorexia nervosa and 18 healthy subjects before and after integrated hospital treatment (nourishment and psychological therapy). The default mode, salience, and frontal-parietal networks were examined using independent component analysis. Body mass index and psychometric measurements significantly improved after treatment. Before treatment, default mode network functional connectivity in the retrosplenial cortex and salience network functional connectivity in the ventral anterior insula and rostral anterior cingulate cortex were decreased in anorexia nervosa patients compared with those in controls. Interpersonal distrust was negatively correlated with salience network functional connectivity in the rostral anterior cingulate cortex. Default mode network functional connectivity in the posterior insula and frontal-parietal network functional connectivity in the angular gyrus were increased in anorexia nervosa patients compared with those in controls. Comparison between pre- and post-treatment images from patients with anorexia nervosa exhibited significant increases in default mode network functional connectivity in the hippocampus and retrosplenial cortex, and salience network functional connectivity in the dorsal anterior insula following treatment. Frontal-parietal network

Only researchers who have formally applied to and been approved by the human research ethics committee of Kyushu University Hospital can access the data (ijkseimei@jimu.kyushu-u.ac.jp).

**Funding:** This work was supported by MEXT KAKENHI Grant Number JP26460910, JP15K08921, and AMED Grant Number JP23dm0307104. The funders had no role in study design, data collection and analysis, decision to publish, or preparation of the manuscript.

**Competing interests:** The authors have declared that no competing interests exist.

functional connectivity in the angular cortex showed no significant changes. The findings revealed that treatment altered the functional connectivity in several parts of default mode and salience networks in patients with anorexia nervosa. These alterations of neural function might be associated with improvement of self-referential processing and coping with sensations of discomfort following treatment for anorexia nervosa.

## Introduction

Anorexia nervosa (AN) is characterized by extremely low body weight, intense fear of weight gain, body image distortion, and extreme weight-control behavior. Individuals suffering from this condition exhibit impaired cognition when self-evaluating their body weight and shape, an inability to recognize the dangers posed by their current low body weight and malnutrition [1], high levels of alexithymia [2], and disturbances in social emotional functioning [3, 4]. Inappropriate self-referential processing, disturbed interoceptive awareness, and excessive cognitive control have been hypothesized to constitutes the basis of the psychopathology of patients with AN [5]. Excessive concentration on body shape and weight can restrict the spheres of life that patients with AN are able to engage in, such as important normative age-graded experiences and introspection, resulting in interpersonal deficits [6].

Understanding the brain function exhibited by patients with AN may be helpful for clarifying the underlying neurobiological mechanisms involved in the disorder. There is a need to increase understanding of the comprehensive neural networks of these psychopathologies in patients with AN. Therefore, we focused on intrinsic network functional connectivity (FC) in resting-state functional magnetic resonance imaging (rsfMRI) analysis, which maps functionally connected brain networks, on the basis of spontaneous non-task related fluctuations of blood oxygen level-dependent (BOLD) signals in the resting brain [7, 8]. Resting-state networks (RSNs) are sets of brain areas exhibiting strong FC (the cross-correlation between BOLD signals in different regions in the resting brain), which play specific functional roles in brain activity at rest [9, 10].

It has been suggested that self-referential emotional processing occurs in the default mode network (DMN) [10], which comprises the precuneus, posterior cingulate cortex (PCC), retrosplenial cortex (RSC), medial prefrontal cortex, lateral temporal cortex, inferior parietal lobule, and hippocampal formation [11], parts of which have been generally related to emotion processing [12]. The DMN exhibits vigorous activity during rest [13]. The DMN is associated with self-processing and self-consciousness, and may thus be relevant to introspection [11, 14, 15].

The DMN is deactivated antagonistically when the frontal-parietal network (FPN) is active, including the dorsolateral prefrontal cortex and parietal cortices [16], which underlie executive functioning, such as working memory and goal-oriented (top-down) cognition [17–19].

Another important network, the salience network (SN), plays a role in switching between the DMN and FPN. The SN consists of the anterior insula (AI) and anterior cingulate cortex (ACC); the former detects salient events of interoceptive and exteroceptive sensation and emotion, whereas the latter facilitates coping with these events [20]. We focused on the DMN and associated networks (i.e., the SN and FPN) to reveal the neural pathology of self-referential function, interoceptive awareness, and cognitive regulation in patients with AN.

In previous studies of AN, analyses of the DMN have reported different results in subjects with AN in different states at different disease stages [21–26]. Individuals with AN in the disease state have been reported to exhibit less DMN FC in the precuneus, whereas individuals

who had recovered from AN exhibited no difference in DMN FC compared with healthy controls (HCs) [25].

Previous studies of SN in patients with AN have also reported different results, possibly depending on the disease stage. Network FC involving the ACC was reported to be reduced in patients with current AN [24, 25] and participants who had recovered from AN [25] compared with that in HCs, whereas some other studies indicated no alteration in the SN FC in participants who had recovered from AN [23].

In contrast, previous studies of the FPN, representing executive cognitive control, have reported consistent results across disease stages in AN, such as higher FC between the FPN and angular gyrus in both AN patients and in patients who had recovered from AN compared with that in HCs [23, 27]. However, all of the studies mentioned above were cross-sectional. In addition, because current and recovered patients were included in different groups in previous studies, it remains unclear whether individual patients exhibit changes in networks according to the progression of the disease state at the within-subject level.

According to the hypothesis that resting-state FC in patients with AN changes with treatment, two previous studies investigated FC in these patients before and after inpatient weight restoration treatment for 2–4 weeks [28, 29]. In adolescents and young adults with AN, using seed-based analysis, FC of the nucleus accumbens with the orbitofrontal cortex was found to be higher in patients with AN before treatment than in HCs, and was decreased after treatment compared with before treatment [28, 29]. FC between networks (SN-FPN/SN-DMN/DMN-FPN) was not affected by the treatment, whereas the connectivity between SN and FPN was reduced in patients with AN relative to HCs [28]. The treatment strategy adopted in these two previous studies, however, was a short-term, inpatient physical treatment focusing on weight gain, raising the question of whether the treatment had a sufficient therapeutic effect on cognitive changes or psychopathology in patients with AN. Furthermore, the analyses relied only on between-network connectivity, which is not always interpretable. Thus, it remains unclear whether each individual network (DMN, SN, and FPN) exhibited functional changes.

The purpose of the current study was to clarify the longitudinal treatment effect of structural cognitive behavioral therapy standardized for AN, called the "cognitive behavioral approach with behavioral limitation," which is a type of reinforcement therapy [30] that is typically conducted for 3–5 months in inpatient treatment. The treatment effect was measured not only using behavioral psychometric scales but also with within-network FC in each of the three different RSNs identified by independent component analyses (ICA) on rsfMRI data. We hypothesized that (1) within-network connectivity in pre-treatment AN patients would be altered in the DMN, SN, and FPN compared with HCs and post-treatment AN patients; (2) these alterations would be associated with individual psychological outcomes related to respective function (self-referential processing, interoception, and cognitive regulation); and (3) pre-treatment AN-specific alterations of RSN would be improved after treatment. Using these combined methods may be helpful for identifying effective treatment targets for AN in the future.

## Materials and methods

### Ethics

The current study was approved by the human research ethics committee at Kyushu University Hospital. Written informed consent for the patients' treatment and these studies was obtained from all participants and from their parents or guardians of minors. All procedures involved in this work complied with the ethical standards of the relevant national and institutional committees on human experimentation, and with the Helsinki Declaration of 1975, as revised in 2008.

## Participants

All participants with AN and HCs were girls or women, right-handed, and aged between 15 and 50 years old. Thirty AN patients consented to participate in this study and received standardized hospital treatment for AN between 2011 and 2015 at our hospital. Patients were diagnosed with AN according to the DSM-IV by clinicians specialized in eating disorders using the Mini-International Neuropsychiatric Interview, and patients with severe depression, suicidal risk, personality disorders, schizophrenia, or alcohol dependence were excluded. Patients with mild or moderate depression, anxiety, or obsessive-compulsive symptoms were included because these symptoms are often comorbid with AN. Patients with AN were allowed to take their required medications. Data were acquired from patients with AN on the first day and the last day of treatment (pre-AN and post-AN, respectively). Eight patients dropped out of the treatment. The data of four patients were excluded from analysis because of poor registration of the fMRI data. Thus, the final longitudinal sample was 18 patients with AN (eight restricting type, 10 binge eating/purging type).

Eighteen HC subjects were recruited from the local community and were required to have no history of eating disorder or other mental illness. HCs were also required not to take any medications. No longitudinal data were collected from HCs.

## Integrated hospital treatment

We treated patients with AN with inpatient therapy called the "cognitive behavioral approach with behavioral limitation" [30]. In this therapeutic approach, patients consented to setting a target body weight and undergoing behavioral limitation (e.g., S1 Table). In addition, patients take part in psychological interviews for their behavioral problems, while they are undergoing nourishment therapy with behavioral limitation. Patients initially received small meals, for easy ingestion. However, if a patient was unable to intake the minimum amount required for nourishment (35 kcal/kg body weight), nasogastric feeding was administered, with patient consent, to compensate for the lack of oral feeding. After confirmation by therapists that a patient was able to ingest the whole meal without difficulty, the amount of the meal was increased gradually by approximately 200 kcal/day, and nasogastric feeding was gradually reduced. Behavioral limitation plays a role in promoting introspection by controlling external stimulation. Gradual removal of behavioral limitations leads to gradual adaptation to real life. S1 Table shows an example of the behavioral limitations and the schedule for lifting them. From the start of the behavioral limitation until the target body weight is reached, the behavioral limitations were lifted step by step as a reward for every kg of weight gain. When a patient reached the target body weight, the next stage of therapy began, providing a rehearsal of real life. In parallel with the therapy, patients received counseling twice a week to learn how to deal with maladjusted behavior, cognition, and emotion. Group therapy and family counseling were also conducted. Through this combined therapy, patients were expected to realize and correct erroneous notions regarding slenderness, eating behavior and interpersonal relationships. They acquired the ability to notice and express their own emotions, and to think about their lives and interpersonal relationships. As necessary, patients were prescribed anti-depressants, anti-anxiety and/or mood stabilizers for mood disturbance and anti-depressants and/or anti-psychotics for severe obsessive behavior.

## Demographic and psychometric measurements

Each participant's body mass index (BMI) was measured, and a subset of participants completed psychometric questionnaires including the self-rating depression scale (SDS) [31] (pre-AN, n = 15; post-AN, n = 14; HC, n = 18), eating disorder inventory (EDI) (subscales: drive

for thinness, bulimia, body dissatisfaction, ineffectiveness, perfectionism, interpersonal distrust, interoceptive awareness, and maturity fears) [32] (pre-AN, n = 17; post-AN, n = 17; HC, n = 18), and the 20-item Toronto alexithymia scale (TAS-20) (subscales: difficulty in identifying feelings, difficulty in describing feelings, and externally oriented thinking) [33, 34] (pre-AN, n = 17; post-AN, n = 17; HC, n = 18). Group comparisons of these participant characteristics and psychometric measurements were conducted on available data using two-sample $t$-tests ("pre-AN $vs$. HC" and "post-AN $vs$. HC"). Paired-sample $t$-tests were used to analyze a longitudinal subset of patients who completed questionnaires at pre- and post- AN ("pre-AN $vs$. post-AN," SDS, n = 12; EDI, n = 16; TAS-20, n = 16) with IBM SPSS Statistics Version 23.0 (IBM SPSS Inc., Chicago, IL, USA).

## Brain image data acquisition

Imaging was performed with a PHILIPS Achieva 3-Tesla scanner (Best, Netherlands). Subjects lay in a supine position, with foam pads fixing the head and earplugs inserted into the ears to reduce head motion and scanner noise. Resting state was defined as when the subject was not engaging in any specific cognitive task during fMRI scanning [35]. During the acquisition of rsfMRI, the subjects were instructed to remain still, relax, close their eyes, and not think anything in particular. Although fMRI was performed at the first procedure with instructions not to fall asleep before the scan, a sleep scale was not used. We obtained the resting-state functional scans using an echo-planar imaging sequence with the following parameters: 32 axial slices, repetition time = 1,793 ms, echo time = 40 ms, fractional anisotropy = 90˚, slice thickness/gap = 3/1 mm, field of view = 210 × 210 mm, resolution = 3 × 3 × 4 mm, and 160 volumes in total (4 minutes 54 seconds). High-resolution three-dimensional magnetization-prepared rapid gradient-echo T1-weighted images were acquired for anatomical localization with the following imaging parameters: 200 sagittal slices, repetition time = 7.0 ms, echo time = 3.2 ms, fractional anisotropy = 9˚, slice thickness/gap = 1/0 mm, field of view = 256 × 240 mm, resolution = 1 × 1 × 1 mm (6 minutes 31.7 seconds). No participants exhibited structural abnormalities during visual inspection.

## Image data preprocessing

Preprocessing of resting-state functional brain images was performed using SPM12 (Statistical Parametric Mapping, Wellcome Trust, UCL, UK). The first 10 volumes of functional images were removed to eliminate the non-equilibrium effects of magnetization. The raw images were converted to the neuroimaging informatics technology initiative (NIFTI) format. Realignment, slice timing correction, and spatial co-registration were performed. According to the exclusion criteria for head motion correction in previous resting state fMRI analyses [36, 37], translational motion parameters were verified to be less than 1 functional voxel. Rotation motion parameters were verified to be less than 2 degrees. Co-registered images were spatially normalized to Montreal Neurological Institute space [38] with a voxel size of $3 \times 3 \times 3$ mm³ using a standard template in SPM12. The normalized images were then smoothed with an 8 mm full-width at half-maximum Gaussian kernel.

## Independent component analysis

RSN consists of brain regions in which neural activities are temporally correlated and considered to be functionally interconnected. To identify RSNs, an ICA was performed to decompose the rsfMRI voxel-by-voxel signals into temporally independent hemodynamic patterns distributed in different regions [39] using the Group ICA fMRI Toolbox, which operates in Matlab. The number of components was estimated using minimum description length criteria [40].

The dimensionality of the preprocessed rsfMRI data from each subject was reduced using principal component analysis. An ICA using the infomax algorithm was then applied to the data [41]. For each subject, this ICA generated a volumetric map for each component (component image), which contains the contribution of each component's time course to the BOLD signal in each voxel. The individual component images were reconstructed (back-reconstruction using the GICA algorithm) and converted to z-values [39], which were fed into subsequent second-level between-subject analyses, as described below.

## Component selection

The spatial distribution of RSNs was identified using a template based on previous studies [42] obtained from 90 functional regions of interest (fROIs, https://findlab.stanford.edu/functional_ROIs.html). The DMN template comprised the medial prefrontal cortex, PCC, RSC, and medial temporal lobe. The SN template was composed of the AI and dorsal ACC. The FPN template included the bilateral parietal cortex and dorsolateral prefrontal cortex (DLPFC). We chose the component with the highest correlation with the respective template mask.

## Mapping network-related connectivity by groups and treatment stages

To map the three different RSNs' connectivity by groups and treatment stages, whole-brain voxel-by-voxel one-sample t-tests were performed on individual z-transformed component images for each selected component in HCs, pre-AN patients, and post-AN patients (significant at a family-wise error [FWE]-corrected peak-level threshold of $p < 0.05$) (SPM-12). These RSN maps were used to create the RSN masks in the following analysis.

## Group and treatment effect on FC in resting-state networks

To specify the brain regions with AN-specific alterations in FC within each of the three different RSNs of interest, individual component images were compared between pre-AN patients and HCs with second-level group analyses using voxel-by-voxel two-sample t-tests (SPM-12). We employed a cluster defined by an FWE-corrected peak-level threshold of $p < 0.05$. To ensure that the differences selectively reflected the FC within the network of interest, these analyses were restricted within the respective RSN masks that were created by overlap between the gray matter mask and the conjunction of network-related connectivities in pre-AN patients and HCs (significant at a FWE-corrected peak-level threshold of $p < 0.05$).

In an exploratory analysis to identify the brain regions exhibiting a treatment effect on FC within each of three different RSNs, individual component images were compared between the pre-AN and post-AN groups in second-level group analyses using voxel-by-voxel paired t-tests (SPM-12). We employed a cluster defined by an FWE-corrected peak-level threshold of $p < 0.05$. To ensure that the differences reflected the FC within the network of interest selectively, these analyses were restricted within the respective RSN masks that were created from overlap between the gray matter mask and the conjunction of network-related connectivities in the pre-AN and post-AN groups (significant at an FWE-corrected peak-level threshold of $p < 0.05$). We used the gray matter mask from WFU PickAtlas (http://fmri.wfubmc.edu/software/PickAtlas).

## Definition of regions of interest

To further investigate the detailed features of the RSN connectivity in brain regions involved in AN pathology, and to determine whether such pathological connectivity would be changed with treatment, we defined the regions of interest (ROIs) that were specific to AN pathology.

The spheres (5 mm radius) centering the peak of significant clusters were detected by comparison between pre-AN patients and HC, and the detected spheres were defined as the ROIs specific to AN pathology. In some cases, regions detected by the comparison were assumed to be small and consisted almost exclusively of peaks. To show that the peaks were not false positives and to correct for this possibility, the regions surrounding the peaks, including the peaks, were defined as the ROI. Next, we calculated the mean of component contribution values within the voxels in ROIs in the individual component image for pre-AN patients, HC, and post-AN patients using Marsbar software [43]. These values represent the strengths of network connectivity to whole brain. The individual mean component values at these ROIs were used for the following two analyses: a multiple regression analysis to identify demographic and psychometric indices related to RSN FC in these ROIs in each of pre-AN patients and HC, and an analysis of treatment effects on RSN FC in these ROIs by comparing the post-AN and pre-AN groups.

### Relationship of RSN FC in ROIs to psychometric measurements

Multiple regression analysis was used to evaluate the demographic and psychometric factors that most strongly influenced differences in RSN FC between AN patients and HCs in AN-specific ROIs. Independent variables were set from indices that were significantly different between pre-AN individuals and HCs in psychometric measurements (SDS, EDI subscales, and TAS-20 subscales). Covariates were set from demographic indices (age, education, BMI, disease duration, and medication use). For a subset of participants who completed all psychometric measurements (i.e., pre-AN, n = 15, and HC, n = 18), we set mean contribution values within the ROIs as the dependent variables, psychometric measurements as independent variables selected with the stepwise method, and demographic indices as covariates with the forced entry method using SPSS.

### Treatment effects on RSN FC within AN-specific ROIs

In addition to the exploratory analysis of the treatment effect on RSN FC, we investigated the treatment effect on network connectivity within the AN-specific ROIs in each of the different networks of interest. The mean values in the ROIs in individual component images were compared between pre-AN and post-AN patients using paired-sample $t$-tests (SPM-12, Marsbar), thresholded at a significance level of $q < 0.05$ with false discovery rate correction for multiple comparisons [44]. Additionally, correlation between a change of RSN FC in the ROIs and increase in BMI by treatment was calculated.

## Results

### Demographic indices and psychometric measurements

There were no significant differences in age between AN patients and HCs. However, HCs had a longer duration of education than patients. The pre-AN group had significantly lower BMI values than both the post-AN and HC groups. The post-AN group had significantly higher BMI values than the pre-AN group, but had significantly lower BMI values than the HC group (Table 1). Although the post-AN group tended to recover body weight loss, they did not recover enough to reach the normal weight level.

In almost all psychometric measurements, the pre-AN group had more pathological characteristics than the HC group (Table 1). The pre-AN group exhibited higher levels of depressive symptoms (SDS), higher ED pathology (EDI total and all eight subscales) and higher alexithymia (TAS-20 total and its two subscales). There was no difference in TAS-20 externally oriented thinking.

**Table 1. Demographic and psychometric characteristics of study participants.**

| Characteristics | Group; mean ± SD | | | t, p value | | |
| | Patients with anorexia nervosa, n = 18 | | Healthy controls, n = 18 | | | |
| | Pre | Post | | Pre vs HC | Post vs HC | Pre vs Post |
|---|---|---|---|---|---|---|
| Age, [range], Y | 28.5 ± 9.9, [15–47] | | 28.4 ± 7.5, [22–44] | 0.19, 0.985 | | |
| BMI, kg/m² | 13.4 ± 1.7 | 16.0 ± 1.2 | 20.2 ± 1.8 | 11.46, < 0.001 | 8.03, < 0.001 | 9.94, < 0.001 |
| Education, Y | 13.2 ± 2.1 | | 15.5 ± 0.5 | 4.46, < 0.001 | | |
| Disease duration, M | 110 ± 109 | | 0 | | | |
| Treatment period, D | | 124 ± 47 | | | | |
| SDS | 52.7 ± 9.1 | 40.6 ± 8.3 | 39.8 ± 8.0 | 4.34, < 0.001 | 0.05, 0.959 | 3.16, 0.009 |
| EDI, Drive for thinness | 8.5 ± 6.3 | 4.5 ± 3.9 | 2.6 ± 4.2 | 3.27, 0.003 | 1.44, 0.160 | 2.52, 0.023 |
| Bulimia | 7.2 ± 7.7 | 1.2 ± 2.1 | 1.7 ± 2.0 | 3.08, 0.006 | 0.06, 0.952 | 3.13, 0.007 |
| Body dissatisfaction | 13.6 ± 3.8 | 9.1 ± 5.0 | 9.1 ± 6.3 | 2.55, 0.015 | 0.00, 0.999 | 3.76, 0.002 |
| Ineffectiveness | 16.5 ± 5.6 | 10.0 ± 6.5 | 4.9 ± 4.4 | 6.86, < 0.001 | 2.68, 0.011 | 4.33, 0.001 |
| Perfectionism | 4.8 ± 4.3 | 3.2 ± 3.4 | 1.9 ± 2.6 | 2.39, 0.023 | 1.31, 0.198 | 2.12, 0.051 |
| Interpersonal distrust | 7.4 ± 3.6 | 5.6 ± 3.7 | 3.0 ± 2.0 | 4.54, < 0.001 | 2.48, 0.021 | 1.94, 0.071 |
| Interoceptive awareness | 12.1 ± 5.5 | 4.1 ± 4.6 | 1.7 ± 2.1 | 7.30, < 0.001 | 1.97, 0.061 | 6.92, < 0.001 |
| Maturity fears | 10.0 ± 5.3 | 8.1 ± 5.7 | 5.4 ± 2.6 | 3.13, 0.005 | 1.76, 0.09 | 2.63, 0.019 |
| Total | 80.0 ± 30.0 | 45.8 ± 24.1 | 29.8 ± 19.4 | 5.91, < 0.001 | 2.16, 0.038 | 5.11, < 0.001 |
| TAS-20, DIF | 23.5 ± 4.7 | 16.7 ± 5.2 | 14.3 ± 5.8 | 5.15, < 0.001 | 1.30, 0.204 | 3.23, 0.003 |
| DDF | 18.5 ± 3.6 | 16.5 ± 3.0 | 13.7 ± 3.8 | 3.80, 0.001 | 2.40, 0.022 | 1.93, 0.063 |
| EOT | 20.2 ± 3.6 | 19.2 ± 3.3 | 18.2 ± 3.5 | 1.67, 0.104 | 0.88, 0.386 | 1.37, 0.180 |
| Total | 62.2 ± 8.3 | 52.4 ± 8.9 | 46.2 ± 10.0 | 5.15, < 0.001 | 1.93, 0.063 | 3.16, 0.003 |
| Medication use, no. | 12 | 12 | 0 | | | |

Pre: anorexia nervosa in pre-treatment, Post: anorexia nervosa in post-treatment, HC: healthy control

SD: standard deviation, BMI: body mass index, SDS: self-rating depression scale, EDI: eating disorder inventory, TAS-20: 20-item Toronto alexithymia scale, DIF: Difficulty identifying feelings, DDF: Difficulty describing feelings, EOT: Externally oriented thinking Pre vs HC, Post vs HC; Two-sample t-tests. Psychometric measurements were analyzed in a subset of participants who completed SDS (Pre, n = 15; Post, n = 14; HC, n = 18), EDI (Pre, n = 17; Post, n = 17; HC, n = 18), TAS-20 (Pre, n = 17; Post, n = 17; HC, n = 18).

Pre vs Post: Paired t-tests. Psychometric measurements were analyzed in a subset of patients who completed the SDS (n = 12), EDI (n = 16) and TAS-20 (n = 16) at both the Pre and Post time points.

The details of medication use are shown in S2 Table.

The treatment effect was shown not only by a gain in body weight, but also by improvement of most psychometric measurements. Post-AN scores were significantly decreased for the SDS, EDI total (subscales: drive for thinness, bulimia, body dissatisfaction, ineffectiveness, interoceptive awareness, and maturity fears), and TAS-20 total (subscale: difficulty in identifying feelings), whereas the treatment showed a trend-level effect on perfectionism, interpersonal distrust, and difficulty describing feelings (DDF).

The results revealed that, after treatment, AN patients approached the level of HCs in some psychological measurements: the post-AN group did not significantly differ from the HC group in SDS scores, EDI scores (subscales: drive for thinness, bulimia, body dissatisfaction, perfectionism, interoceptive awareness, and maturity fears), and TAS-20 total scores (difficulty in identifying feelings). However, significant differences in EDI ineffectiveness and EDI total remained between the post-AN and HC groups, despite a significant treatment effect. Significant differences in EDI interpersonal distrust and TAS-20 DDF also remained between the post-AN and HC groups, and a significant treatment effect was not found. This indicates that post-treatment AN patients still exhibited AN-related pathological tendencies for some psychological features.

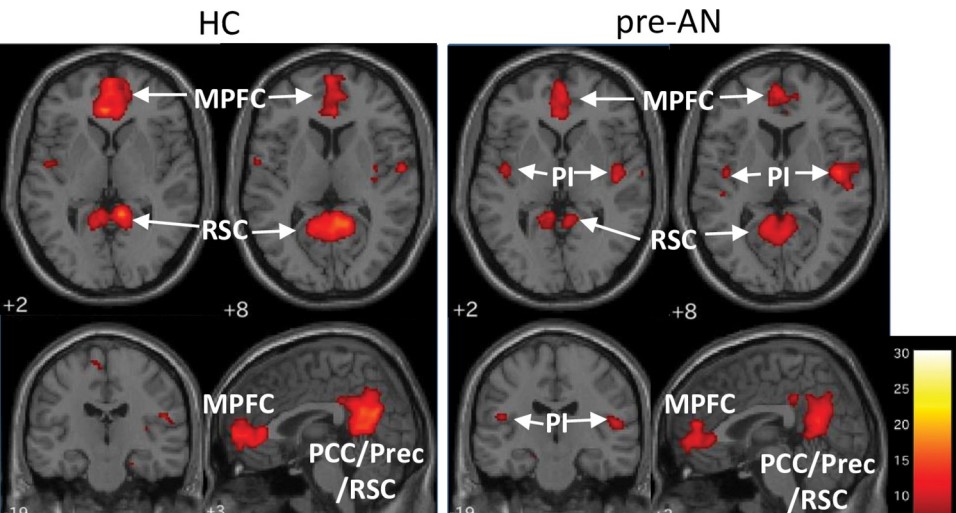

**Fig 1. Default mode network FC maps in HC and pre-AN.** Spatial maps are plotted as t statistics thresholded at p < 0.05 and are family-wise error-corrected. MPFC: medial prefrontal cortex, PI: posterior insula, RSC: retrosplenial cortex, PCC: posterior cingulate cortex, Prec: precuneus. See S1 Fig for post-AN DMN and other RSNs.

## Component identification and RSN FC maps by groups and treatment stages

ICA extracted 17 independent components, among which three components were identified as RSNs of interest (correlation with the network template, component 12 and DMN: $r = 0.473$, component 16 and SN: $r = 0.388$, component 3 and FPN: $r = 0.510$). RSN FC maps of interest are shown for each group (Fig 1 and S1 Fig). The DMN included clusters in the medial prefrontal cortex, PCC/precuneus/RSC, posterior insula (PI)/transverse temporal gyrus and hippocampus. The PI is not generally considered a major region of the DMN. However, in our study, especially in AN, clusters in the PI were detected as part of the DMN (Fig 1 and S1 Fig). Clusters in the ACC and AI were observed in the SN (S1 Fig). Clusters in the right DLPFC and left angular gyrus (AG) were observed in the FPN (S1 Fig).

## Exploratory analyses of group differences and treatment effects on FC in RSNs

Comparison between the pre-AN and HC groups revealed AN-specific alterations in the main regions of DMN, SN, and FPN. The pre-AN group showed significantly decreased DMN FC compared with HCs in the RSC (Table 2 and Fig 2A). In the SN, the pre-AN group exhibited significantly decreased FC compared with HCs in the ventral AI (vAI) and rostral ACC (rACC) (Table 2 and Fig 2B, 2C). The pre-AN group showed significantly higher DMN FC than HCs in the PI (Table 2 and Fig 2D), although PI is generally not the main region of DMN. In the FPN, pre-AN exhibited significantly increased FC compared with HC in the AG (Table 2 and Fig 2E).

We found a treatment effect in a few regions within the DMN and SN by comparison of FC maps of interest between the pre-AN and post-AN patients. The post-AN group showed significantly increased DMN FC in the hippocampus (Table 2 and Fig 3A) and SN FC in the dorsal AI (dAI) (Table 2 and Fig 3B) compared with the pre-AN group. No treatment effect was observed in the FPN. We did not identify any regions with FC that was significantly decreased by the treatment.

**Table 2. Group difference and treatment effect on FC in resting-state networks.**

| Comparison; Network | Region | Vol. | T<sub>max</sub> | Peak coordinates (x,y,z) |
|---|---|---|---|---|
| HC > pre-AN | | | | |
| Default mode network | Retrosplenial cortex | 1 | 4.51 | 3, −55, 8 |
| Salience network | Ventral anterior insula | 2 | 4.59 | −30, 14, −13 |
| | Rostral anterior cingulate cortex | 1 | 4.92 | 0, 41, −7 |
| pre-AN > HC | | | | |
| Default mode network | Posterior insula | 2 | 4.69 | 45, −19, 2 |
| Frontal-parietal network | Angular gyrus | 11 | 5.44 | −39, −73, 35 |
| post-AN > pre-AN | | | | |
| Default mode network | Hippocampus | 1 | 5.37 | 27, −19, −16 |
| Salience network | Dorsal anterior insula | 1 | 5.53 | −27, 17, 11 |
| pre-AN > post-AN | | | | |
| | No suprathreshold clusters | | | |

Peak-level threshold $p < 0.05$, family-wise error-corrected

HC: healthy control, pre-AN: anorexia nervosa patient in pre-treatment

post-AN: anorexia nervosa patient in post-treatment

### Relationship of RSN FC in AN-specific ROIs with psychometric measurements

SN FC in the rACC ROI was positively correlated with education ($\beta = 0.441$, $p = 0.036$), and negatively correlated with EDI interpersonal distrust ($\beta = -0.489$, $p = 0.008$). FPN FC in the AG ROI was positively correlated with TAS-20 DDF ($\beta = 0.354$, p = 0.034). In the other ROIs, resting-state FC was not significantly related to demographic indices or psychometric measurements.

### Pre- and post-treatment comparison of RSN FC in AN-specific ROIs

In addition to the exploratory analysis of treatment effects on RSN FC, we examined the treatment effects on resting-state FC in the AN-specific ROIs in each of the DMN, SN, and FPN. The group comparison of post-AN versus pre-AN exhibited significantly increased FC in the RSC within the DMN (Fig 4A). Treatment effects on the FC were observed as increased connectivity also in the vAI and rACC within the SN, and in the PI within the DMN, although these effects did not remain statistically significant after multiple comparison correction (Fig 4B–4D). Thus, the treatment did not lead to significant improvement of FPN FC in the AG (Fig 4E). A change of RSN FC in any ROI was not significantly correlated with an increase of BMI.

### Discussion

The current study revealed several main findings, as follows: 1) AN-specific alterations in brain regions within the DMN, SN, and FPN (e.g., the RSC within the DMN, vAI, and the rACC within the SN) that exhibited lower FC in the pre-AN group compared with the HC group. The PI within the DMN and AG within the FPN exhibited high FC in pre-AN in comparison with HC. 2) The SN FC in the rACC ROI was negatively correlated with EDI interpersonal distrust. FPN FC in the AG ROI was positively correlated with TAS-20 DDF. 3) Exploratory analyses for the treatment effect (post-AN versus pre-AN) exhibited increased FC in the hippocampus within DMN and dAI within the SN. When we focused on AN-specific

## HC > pre-AN

### Default mode network

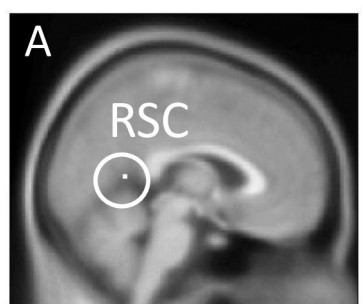

### Salience network

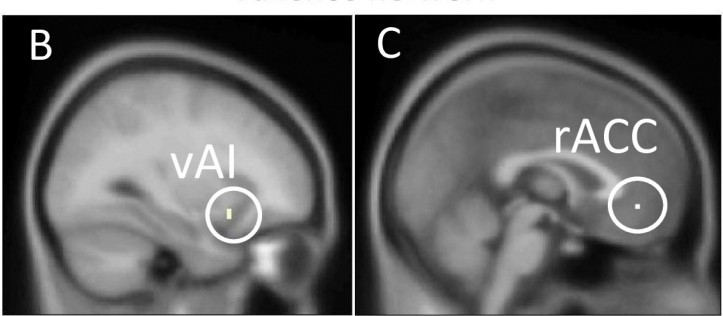

## pre-AN > HC

### Default mode network

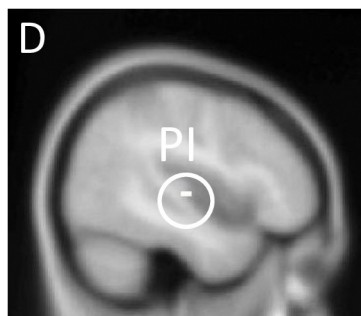

### Frontal-parietal network

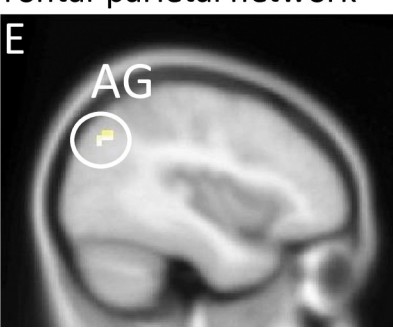

**Fig 2. Group difference of FC in resting-state networks.** The pre-AN group exhibited less connectivity than HCs between (A) the DMN and the retrosplenial cortex (RSC) (peak coordinate: 3, −55, 8), (B) the SN and the ventral anterior insula (vAI) (−30, 14, −13), and (C) the SN and the rostral anterior cingulate cortex (rACC) (0, 41, −7). The pre-AN group showed higher connectivity than HCs between (D) the DMN and the posterior insula (PI) (45, −19, 2), and (E) the FPN and the angular gyrus (AG) (−39, −73, 35), Two-sample *t*-test. Peak-level threshold $p < 0.05$ family-wise error-corrected.

## post-AN > pre-AN

### Default mode network

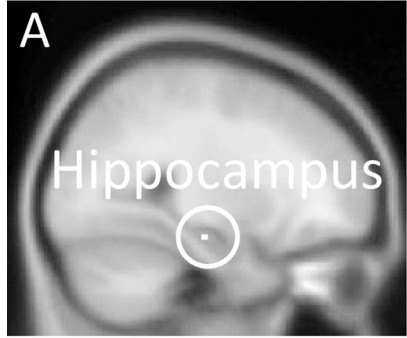

### Salience network

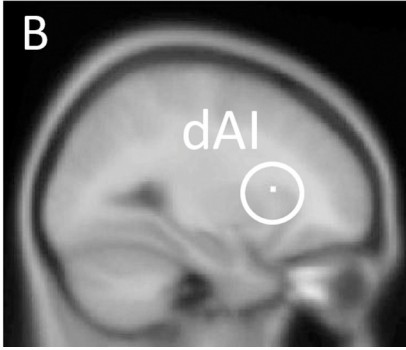

**Fig 3. Treatment effect on FC in resting-state networks.** The post-AN group showed higher connectivity compared with the pre-AN group between (A) the DMN and the hippocampus (27, −19, −16), and (B) the SN and the dorsal anterior insula (dAI) (−27, 17, 11), Paired t-test. Peak-level threshold $p < 0.05$ family-wise error-corrected.

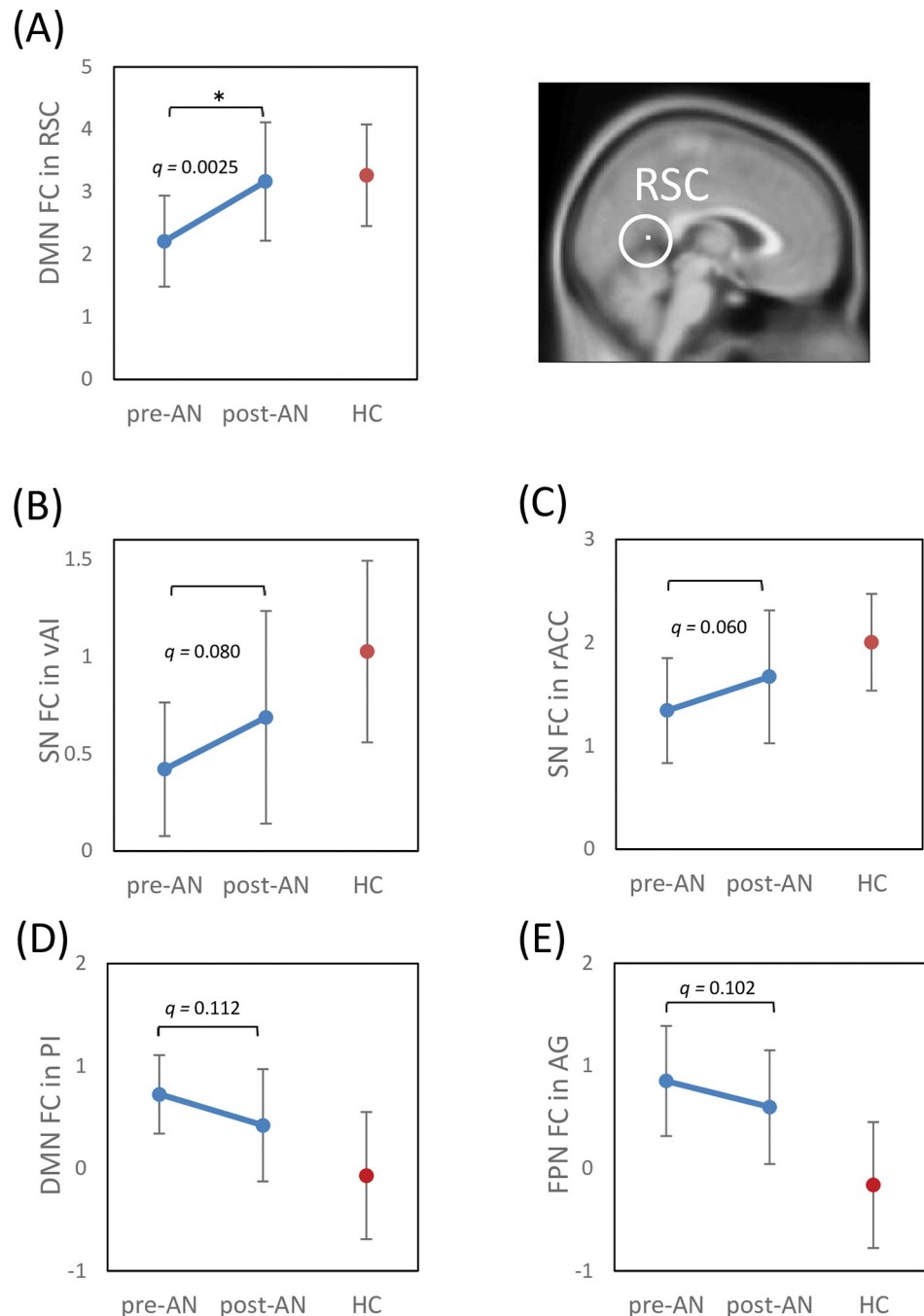

**Fig 4. Comparison of FC of AN-specific ROIs in the pre- and post-treatment images from AN patients (with HCs as a reference).** (A) Default mode network, retrosplenial cortex. (B) Salience network, ventral anterior insula. (C) Salience network, rostral anterior cingulate cortex. (D) Default mode network, posterior insula. (E) Frontal-parietal network, angular gyrus. ROI analysis, Paired $t$-test using Marsbar. * $q < 0.05$, false discovery rate correction for multiple comparisons.

ROIs, the treatment effect was shown as increased FC in the RSC within DMN. Our main hypotheses were supported, as follows: (1) altered within-network connectivity was observed in pre-treatment AN patients. (2) There were relationships between these alterations and individual psychological outcomes. (3) We observed a treatment effect on RSNs.

First, our findings indicated that DMN FC in the RSC showed AN-specific changes and treatment effects. The RSC is known to be structurally and functionally connected to the hippocampus and plays a role in self-reference processing while accessing memory information, such as retrieval of episodic memory, autobiographical memory, navigation with imagination, thinking about the future, introspection, and theory of mind [45–50]. Functional alterations in the RSC have been reported in psychiatric disorders involving impaired self-referential function, such as schizophrenia [51], bipolar disorder [52], post-traumatic stress disorder [53, 54], social anhedonia [55, 56], individuals with high-trait-anxiety [57], and autism [58]. The current results are consistent with a previous study of AN reporting lower DMN FC than HCs in the region containing the RSC with no significant differences between HC and recovered patients [25]. As our behavioral results also revealed a significant recovery in self-cognition function after treatment, low DMN FC in the RSC suggests that impaired self-cognition is a specific feature in AN patients with symptoms. Thus, clinical improvement might be associated with amelioration of DMN FC in the RSC.

The significant recovery of DMN FC in the RSC with our integrated treatment might depend on either nourishment therapy or psychotherapy, or both. This finding could not be explained by a simple correlation with weight gain. Previous studies reported that DMN FC was strengthened by psychotherapy. For example, cognitive behavioral therapy for patients with chronic pain increased the amplitude of low-frequency fluctuation in the cerebellum and PCC (close to the RSC, which is part of the DMN) and FC between these areas [59]. A focused attention meditation increased connectivity from the striatum to the PCC and RSC [60]. Not only recovery of body weight owing to nourishment but also the psychotherapy in our program appeared to exert a therapeutic effect on DMN FC.

With an exploratory analysis, we found that DMN FC in the hippocampus was increased following treatment, even though it was not detected as AN-specific alteration in DMN FC. Hippocampal functional improvement might be associated with progress of introspection in patients with AN. Generally, the hippocampus is coupled with the DMN during memory retrieval [61], and plays an important role in emotional regulation [62]. The hippocampus is known to be structurally and functionally connected to the RSC [46, 47], so that the treatment-induced increase of DMN FC in the hippocampus observed in our study is considered to be related to improvement in the RSC. Several studies have reported that psychotherapy, including cognitive behavioral therapy, increased hippocampal function in patients with other psychological disorders [63], such as major depression [64] and post-traumatic stress disorder [65]. The increased DMN FC in the hippocampus observed in the current study might be also associated with improvement of neural function by repeated introspection in psychotherapy, which is coupled with the improvement of DMN FC in the RSC.

The exploratory analysis also revealed that the SN FC in the dAI was increased following the treatment, despite the absence of AN-specific changes at baseline. The activity or FC in dAI has been reported to change after treatment for other diseases, such as schizophrenia [66] and major depressive disorder [67], suggesting that the SN FC change in the dAI observed in our study might not be specific to AN. The dAI is involved both in processing somatosensory inputs and decision-making in the initiation of behavior. This area integrates internal and external sensory information to coordinate brain network dynamics to initiate switches of the DMN and FPN [68–70], leading to behavioral changes. Improved SN FC in the dAI might promote appropriate behavior in response to perceived physical or psychological discomfort, such as perception of hunger in AN, which could be associated with improvement of the symptoms of AN.

The right PI revealed higher FC to DMN in pre-AN patients than in HCs. The interaction between the DMN and PI might be associated with abnormal somatic sensation in patients

with AN. The PI would be associated with pain and somatosensory processes [69, 71]. Patients with chronic low back pain showed high FC between the DMN and PI [72], which is regarded as a diseased phenomenon. Patients with AN also present bodily complaints frequently, so their high FC in the PI to the DMN might indicate the psychological modification of somatosensory information [73], which relates to their mind-induced somatic symptoms.

The vAI showed lower SN FC in pre-AN patients compared with that in HCs. These phenomena might be associated with impaired socio-emotional processing, such as emotional cognition and empathy in patients with AN [74–77]. The vAI is connected to the rostral ACC [70], which plays an important role in socio-emotional processing in the SN [20, 69, 78, 79]. Abnormal function in the vAI has been reported in other diseases involving socio-emotional function, including bipolar disorder [80] and bronchial asthma with depression [81]. SN FC of the rACC was lower in pre-AN patients compared with that in HCs, and was negatively correlated with interpersonal distrust. These findings have important implications for social maladjustment in patients with AN. The rACC is involved in empathy [82–84], and dysfunction in this region is believed to be related to maladaptive emotional processing and interpersonal stress [85, 86]. Hypofunction of the rACC was reported in several previous resting-state neuroimaging studies of patients with AN [21, 24, 87].

The current study showed that FPN FC in the AG was higher in pre-AN patients than in HCs and was positively correlated with TAS-20 DDF. The FPN is associated with executive cognition control [17–19], which is reported to be excessive in patients with AN from a clinical perspective [27, 88–90]. Patients with AN have been found to exhibit predominant FPN activity during set shifting and cognitive flexibility [88, 91], which may reflect that top-down cognitive control is dominant [89, 90, 92]. This excessive functioning might be an impediment to expression of fluid feelings. Previous resting-state network studies in both patients with AN and participants who had recovered from AN reported higher FPN FC in the AG [23, 27]. In the current study, post-AN patients showed no significant improvement. This neural function was not changed by the treatment, potentially indicating a neural trait in patients with AN.

Taken together, our findings indicated that DMN FC in the RSC, which is involved in self-reference and coping, showed significant changes with treatment, suggesting that this element is more "improvable." This may have occurred because, among the various elements of our psychotherapeutic inpatient treatment, the promotion of introspection was reflected by improvements in neurological functioning. It is generally considered that treatments focused on improvable function are more likely to be effective. This view suggests that currently used treatments (i.e., enhanced cognitive behavior therapy, Maudsley model therapy, and focal psychodynamic psychotherapy) contain elements of introspection, compensate for weak functions, and have credible therapeutic effects [6, 93, 94], and should be noted when future treatment is revised.

## Limitations

First, the small sample size is a potential limitation of the current study, which may affect the generalizability of the results. Second, because we adopted an integrated treatment regimen involving nourishment, psychotherapy, and medical treatment, we were unable to attribute the observed effects to a specific elemental therapy. Psychotherapy, as well as changes in body weight and medication use, may have affected the change in FC. Third, many patients were mildly relieved by treatment, but had not fully recovered at the time of discharge. Fourth, we were unable to analyze how the neural change with treatment influenced the clinical course after discharge. Future studies with a larger sample size may be needed to investigate the improvement in RSN FC with therapy, differences in RSN FC depending on the subtypes or clinical characteristics, and the influence of treatment on long-term clinical consequences.

## Conclusion

We acquired rsfMRI from patients with AN before and after undergoing integrated inpatient treatment. We analyzed RSNs of interest using ICA. Post-AN patients exhibited higher DMN FCs in the RSC and the hippocampus than pre-AN patients. These results might be related to the improvement of self-referential function and progress of introspection induced by the integrated treatment. Post-AN patients exhibited higher SN FC in the dAI than pre-AN patients. The increase in FC following treatment might promote appropriate coping with discomfort in emotion and sensation. The RSN FCs in AN-specific ROIs (the right PI in DMN, vAI/rACC in SN and AG in FPN), except RSC in DMN, did not show significant changes between before and after treatment. These phenomena might reflect trait neural pathology regarding abnormal somatic perception, difficulty in socio-emotional processing, and excessive cognitive control/difficulty in describing feelings in patients with AN. The current findings help to elucidate pathology and treatment effects in RSN in patients with AN, which may play a critical role in setting targets for future treatment.

## Supporting information

**S1 Table. An example of behavioral limitation.**
(PDF)

**S2 Table. Details of medication use.**
(PDF)

**S1 Fig. Resting-state network FC maps of interest in each group.**
(PDF)

## Acknowledgments

We thank all of the participants, the nursing staff who engaged in treatment, and the radiological technician.

## Author Contributions

**Conceptualization:** Motoharu Gondo, Yoshiya Moriguchi.

**Data curation:** Motoharu Gondo.

**Formal analysis:** Motoharu Gondo.

**Funding acquisition:** Motoharu Gondo, Keisuke Kawai, Kazufumi Yoshihara.

**Investigation:** Motoharu Gondo, Chihiro Morita, Makoto Yamashita, Sanami Eto.

**Methodology:** Motoharu Gondo, Yoshiya Moriguchi, Akio Hiwatashi.

**Project administration:** Keisuke Kawai, Nobuyuki Sudo.

**Resources:** Motoharu Gondo, Keisuke Kawai, Akio Hiwatashi, Shu Takakura, Chihiro Morita, Makoto Yamashita, Sanami Eto.

**Software:** Motoharu Gondo.

**Supervision:** Yoshiya Moriguchi.

**Validation:** Keisuke Kawai, Nobuyuki Sudo.

**Visualization:** Motoharu Gondo.

**Writing – original draft:** Motoharu Gondo.

**Writing – review & editing:** Yoshiya Moriguchi, Akio Hiwatashi, Shu Takakura, Kazufumi Yoshihara.

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
