## [Decision Letter · Decision Letter 0]

31 Oct 2022

PONE-D-22-26079Effects of integrated hospital treatment on the default mode, salience, and frontal-parietal networks in anorexia nervosa: A longitudinal resting-state functional magnetic resonance imaging studyPLOS ONE

Dear Dr. Gondo,

Thank you for submitting your manuscript to PLOS ONE. After careful consideration, we feel that it has merit but does not fully meet PLOS ONE’s publication criteria as it currently stands. Therefore, we invite you to submit a revised version of the manuscript that addresses the points raised during the review process.

 The two reviewers addressed several major and minor concerns about your manuscript. Please revise your manuscript according to reviewer's comments.

We look forward to receiving your revised manuscript.

Kind regards,

Kenji Hashimoto, PhD

Section Editor

PLOS ONE

Journal Requirements:

Reviewers' comments:

Reviewer's Responses to Questions

**Comments to the Author**

1. Is the manuscript technically sound, and do the data support the conclusions?

Reviewer #1: Yes

Reviewer #2: Partly

2. Has the statistical analysis been performed appropriately and rigorously? 

Reviewer #1: Yes

Reviewer #2: No

3. Have the authors made all data underlying the findings in their manuscript fully available?

Reviewer #1: No

Reviewer #2: Yes

4. Is the manuscript presented in an intelligible fashion and written in standard English?

Reviewer #1: Yes

Reviewer #2: Yes

5. Review Comments to the Author

Reviewer #1: This study investigated abnormalities of the resting-state functional connectivity in the default mode network, salience, and frontal-parietal networks in the patients with anorexia nervosa. The functional connectivity in these three networks were evaluated cross-sectionally and longitudinally. This is very interesting study, there are however some issues to be mentioned.

1) The authors conducted many analyses and obtained many results. However, the manuscript seems to be redundant.

2) In the Ethics section, the authors wrote ‘The current study was approved by the human research ethics committee at our hospital.’ What was ‘our hospital’?

3) In addition, the authors wrote ‘Written informed consent was obtained from all participants.’ Was this informed consent for the neuroimaging studies or/and cognitive behavioral approach with behavioral limitation?

4) In the Participants section, the authors wrote ‘All AN and HC participants were right-handed and aged between 15 and 50 years old.’ However, there was no description about parental consent for the adolescent participants in the Ethics section. This is a critical issue.

5) ‘patients with severe depression, suicidal risk, personality disorders, schizophrenia, or alcohol dependence were excluded’ in this study. How about the obsessive-compulsive disorder that often comorbid with anorexia nervosa?

6) How many numbers of male and female participants are there in each of the AN and HC group?

7) The author should write age range in both the AN and HC groups in the Table2.

8) Information of benzodiazepine should be written in the Table2.

9) I could not understand the meaning of ‘AN-specific changes’ that the authors used. What the ‘AN-specific’ mean in this manuscript?

10) Even though the AN patients received treatments, their mean BMI was 16.0, still small, and the mean total score of the EDI was 45.8, still high. Severity of the AN was changed from severe to mild, but the patients were not recovered. This is one of limitation in this study.

Reviewer #2: Remarks to the authors:

In the present study, the authors measured resting-state functional magnetic resonance images (MRi) from 18 patients with anorexia nervosa (AN) and 18 healthy subjects before and after integrated hospital treatment to examine hypotheses that the resting-state brain networks might be altered in such patients, and that treatment might normalize neural functional connectivity. (nourishment and psychological therapy). They showed that AN of post intervention exhibited higher functional connectivity (FC) in default mode network (DMN) and also higher FC in salience network (SN). The authors concluded results might reflect trait neural pathology regarding abnormal somatic perception, difficulty in socio-emotional processing, and excessive cognitive control in patients with AN. The manuscript is well written, and proposes interesting results in general. What follows are critical concerns and some suggestions to improve the manuscript.

Major Concerns:

1. It is strongly required for the authors to describe adopted denoising methods of their raw imaging data including motion correction and its exclusion criterion, which is currently obligatory for research of resting state functional MRI.

2. It is not popular to define the posterior insula as a part of default mode network, although these misleading findings could be found in results of an exploratory analysis of brain functional networks. It is highly recommended for the authors to interpret results based on understanding shared with those who are expertized in MRI studies, and to adopt appropriate expression to avoid misunderstanding of readers.

Minor Concerns

1. The study design of the current study cannot also distinguish influence of intervention from change of body weight, in the other words malnourished state. The authors should discuss the point in the limitation section.

2. In the introduction part (page 5, line 104), the authors stated that the previous studies showed inconsistent results. However, it is understandable that remitted AN exhibits different state from current AN (ie. ‘State or trait’ issue).

3. The authors should refer reasons why they adopt random effect model and its details, possibly with some previous references.

4. If some “sleep scale” was measured after resting state functional MRI to confirm participants were awake, it should be referred in the method section.

5. The authors should precisely explain what component contribution values are (page 15, line 315). Do they stand for the strengths of within-network connectivity, or those of network connectivity to whole brain?

6. The authors stated that comparison of SDS between pre- and post-intervention was performed among 12 AN participants, but in the section “Relationship of RSN FC in ROIs to psychometric measurements” of page 15, the number of participants is 15 there.

7. As for AN-specific ROIs, it is preferable for authors to carefully describe why they used the sphere mask from the peak voxel, instead of the mask of the region detected in the previous analyses.

8. Just out of my curiosity, it would be welcomed if the authors show correlation between FC and body weight change, if possible. If no correlation was detected in that, it could be indirectly assumed that the FC change between pre- and post- was related to some effect of intervention.

6. PLOS authors have the option to publish the peer review history of their article (what does this mean?). If published, this will include your full peer review and any attached files.

Reviewer #1: No

Reviewer #2: No

---

## [Author Response · Author response to Decision Letter 0]

29 Dec 2022

Thank you for inviting us to submit a revised draft of our manuscript entitled, “Effects of integrated hospital treatment on the default mode, salience, and frontal-parietal networks in anorexia nervosa: A longitudinal resting-state functional magnetic resonance imaging study” to PLOS ONE. We also appreciate the time and effort you and each of the reviewers have dedicated to providing insightful feedback on ways to strengthen our paper. Thus, it is with great pleasure that we resubmit our article for further consideration. We have incorporated changes that reflect the detailed suggestions you have graciously provided. We also hope that our edits and the responses (cover letter and response to reviewers) satisfactorily address all the issues and concerns you and the reviewers have noted.

---

## [Decision Letter · Decision Letter 1]

19 Jan 2023

PONE-D-22-26079R1Effects of integrated hospital treatment on the default mode, salience, and frontal-parietal networks in anorexia nervosa: A longitudinal resting-state functional magnetic resonance imaging studyPLOS ONE

Dear Dr. Gondo,

Thank you for submitting your manuscript to PLOS ONE. After careful consideration, we feel that it has merit but does not fully meet PLOS ONE’s publication criteria as it currently stands. Therefore, we invite you to submit a revised version of the manuscript that addresses the points raised during the review process.

The reviewer #2 addressed additional comments about your manuscript. Please revise your manuscript again. I will make the final decision after you submit to the journal.

We look forward to receiving your revised manuscript.

Kind regards,

Kenji Hashimoto, PhD

Section Editor

PLOS ONE

Reviewers' comments:

Reviewer's Responses to Questions

**Comments to the Author**

1. If the authors have adequately addressed your comments raised in a previous round of review and you feel that this manuscript is now acceptable for publication, you may indicate that here to bypass the “Comments to the Author” section, enter your conflict of interest statement in the “Confidential to Editor” section, and submit your "Accept" recommendation.

Reviewer #1: All comments have been addressed

Reviewer #2: (No Response)

2. Is the manuscript technically sound, and do the data support the conclusions?

Reviewer #1: Yes

Reviewer #2: Partly

3. Has the statistical analysis been performed appropriately and rigorously? 

Reviewer #1: Yes

Reviewer #2: Yes

4. Have the authors made all data underlying the findings in their manuscript fully available?

Reviewer #1: Yes

Reviewer #2: No

5. Is the manuscript presented in an intelligible fashion and written in standard English?

Reviewer #1: Yes

Reviewer #2: Yes

6. Review Comments to the Author

Reviewer #1: This study investigated abnormalities of the resting-state functional connectivity in the default mode network, salience, and frontal-parietal networks in the patients with anorexia nervosa. The functional connectivity in these three networks were evaluated cross-sectionally and longitudinally. This is a valuable study. In the revised manuscripts, the authors have adequately addressed your comments.

Reviewer #2: Remarks to the authors:

The authors have precisely approached all my comments and questions, and the manuscript has been considerably improved. What follows are suggestion to improve the manuscript.

Concerns:

1. In page 19 line 308, it is not assumed that the posterior insula might be detected in the group comparison, if the mask of default mode network (DMN) were appropriately applied. Careful interpretation and handling of this point would be recommended.

2. In page 19 line 316, it should be clarified whether a repeated measures ANOVA was adopted (recommended).

3. In page 20 line 332, the use of sphere-shaped ROI has the risk that regions with different brain functions can be included. The authors should consider to cite an appropriate reference that shows validity of the method in analyzing resting state functional MRI instead they use labels that include peak voxels.

4. In page 20 line 335-337, were mean values of the spheres calculated in these analyses?

5. In page 22 line 364, just asking for confirmation purpose, was multiple comparison correction performed for 5 ROIs?

6. In page 27, the authors mentioned that posterior insula was detected especially in AN. Does it mean that it was visually obvious? It is a bit difficult for readers to understand, and the author should explain its implication.

7. Still concerns in handling of posterior insula as if a part of DMN. The related expression in the discussion section should be carefully amended.

7. PLOS authors have the option to publish the peer review history of their article (what does this mean?). If published, this will include your full peer review and any attached files.

Reviewer #1: No

Reviewer #2: No

---

## [Author Response · Author response to Decision Letter 1]

4 Mar 2023

The followings are the point-by-point responses to the questions and comments that were delivered in the letter dated January 19th.

Reviewer #2: Remarks to the authors:

The authors have precisely approached all my comments and questions, and the manuscript has been considerably improved. What follows are suggestion to improve the manuscript.

Concerns:

1. In page 19 line 308, it is not assumed that the posterior insula might be detected in the group comparison, if the mask of default mode network (DMN) were appropriately applied. Careful interpretation and handling of this point would be recommended.

7. Still concerns in handling of posterior insula as if a part of DMN. The related expression in the discussion section should be carefully amended.

RESPONSE to the questions 1) and 7): In this study, the mask for the DMN was created in a data-driven fashion based on the ICA of the data obtained from current sample, in which the mask was chosen from one of the components from ICA that fitted best to the DMN a priori template. The reason why we did not use the a priori DMN template directly was that areas that function as default mode are not always constant as one fixed structural set of regions but may normally vary depending on the occasion to some extent – especially the characteristics of sample. Also the network functions and interactions within the networks are not stable, but rather, exhibit substantial variability across time and context (Dixon, et al. Interactions between the default network and dorsal attention network vary across default subsystems, time, and cognitive states. NeuroImage. 2017; 147:632-49). Our approach, creating the DMN mask specific to our study sample in a data-driven fashion, is to increase the sensitivity to detect more relevant regions that “actually functioned” as the default mode network, and has been used widely in the past studies and assumed valid and appropriate. 

Through such a ROI setting process, there may be cases where the regions that have not been considered are detected as in the aimed network but closely connected to that network are selected as a part of the data-driven mask, such as the posterior insula (PI) relevant to the DMN. Even though the PI had not been assumed as a part of the DMN, the fact that the PI was selected as a part of the DMN mask indicates that the PI functioned as default mode in our sample (particularly in AN patients) and should be considered as a ROI in the subsequent analyses specifically in our study because it may include important information relevant to the AN pathology.

Nevertheless, according to the reviewer’s suggestion to avoid confusing the readers, we amended the description about the PI in the discussion where the PI had been referred to as if a part of the DMN:

(Page 32, line 556-573)

 The right PI showed higher FC to the DMN FC in pre-AN patients than in HC. These phenomenaThe interaction between the DMN and PI might be associated with abnormal somatic sensation in patients with AN. The PI would be associated with pain and somatosensory processes [67, 69]. The patients with chronic low back pain showed high FC between the DMN and PI [70], which is regarded as a diseased phenomenon. AN patients also present bodily complaints frequently, so their high FC in the PI to the DMN FC in the PI might indicate the psychological modification of somatosensory information [71], which relates to their mind-induced somatic symptoms. 

2. In page 19 line 316, it should be clarified whether a repeated measures ANOVA was adopted (recommended).

RESPONSE: We used paired-sample t-tests for a longitudinal subset of patients who completed questionnaires and scanning at pre- and post-AN as already described in the Method section in the manuscript, if we did not use covariations in the estimation model. Paired sample t-test is statistically equivalent to repeated measures ANOVA to compare changes over time at two time points. 

3. In page 20 line 332, the use of sphere-shaped ROI has the risk that regions with different brain functions can be included. The authors should consider to cite an appropriate reference that shows validity of the method in analyzing resting state functional MRI instead they use labels that include peak voxels.

RESPONSE: The reason why we adopted the sphere ROIs centering the peak voxel was that only selecting the peak voxel as the ROI has the risk to pick up noise signals, so that some smoothing method is needed to denoise the information (i.e., adding the spatial widening like changing from peak to sphere). It has also a risk to use areas according to structural labels because areas only according to structural labels are too broad in many cases and do not necessarily reflect functional entity of the peak voxel in the analysis of time series BOLD signals in rsfMRI. Sphere-shaped functional ROI setting is a popular, conventional and safe method used in functional analysis of the brain and have been frequently used in recent rsfMRI studies [Spets, et al. 2021] [Pahapill, et al. 2020] [Martial, et al. 2019] [Chiang, et al. 2018] [Chaddock-Heyman, et al. 2018]. We adopted 5mm-radius sphere, which is small enough to avoid including functionally irrelevant brain regions outside the detected peaks. 

4. In page 20 line 335-337, were mean values of the spheres calculated in these analyses?

RESPONSE: Yes, we calculated the mean of component contribution values within the voxels in ROIs in the individual component images. We clarify this in the manuscript.

(Page 18, line 312-314)

Next, we calculated the mean of component contribution values within the voxels in ROIs in the individual component image for pre-AN patients, HC, and post-AN patients using Marsbar software [41].

5. In page 22 line 364, just asking for confirmation purpose, was multiple comparison correction performed for 5 ROIs?

RESPONSE: Yes, the data from the 5 ROIs were analyzed with false discovery rate correction for multiple comparisons (page 19 line 336-342).

6. In page 27, the authors mentioned that posterior insula was detected especially in AN. Does it mean that it was visually obvious? It is a bit difficult for readers to understand, and the author should explain its implication.

RESPONSE: Yes, it is visually obvious. We had moved three RSNs in three group (HC, pre-AN, and post-AN) to supporting information (S1 Fig) in the last revision due to the heavy amount of information. However, we have displayed the DMN for HC and pre-AN in Fig 1 for the reader's convenience in the current manuscript.

Lastly, we would like to correct an error in the last manuscript revision in response to a minor concern raised by the reviewer #2, about what component contribution values are: the strengths of within-network connectivity, or those of network connectivity to whole brain. 

(Page 18, line 312-315)

Next, we calculated the mean of component contribution values within the voxels in ROIs in the individual component image for pre-AN patients, HC, and post-AN patients using Marsbar software [41]. These values represent the strengths of within-network connectivity network connectivity to whole brain.

---

## [Editor Report · Decision Letter 2]

8 Mar 2023

Effects of integrated hospital treatment on the default mode, salience, and frontal-parietal networks in anorexia nervosa: A longitudinal resting-state functional magnetic resonance imaging study

PONE-D-22-26079R2

Dear Dr. Gondo,

We’re pleased to inform you that your manuscript has been judged scientifically suitable for publication and will be formally accepted for publication once it meets all outstanding technical requirements.

Kind regards,

Kenji Hashimoto, PhD

Section Editor

PLOS ONE
---

## [Editor Report · Acceptance letter]

19 May 2023

PONE-D-22-26079R2 

Effects of integrated hospital treatment on the default mode, salience, and frontal-parietal networks in anorexia nervosa: A longitudinal resting-state functional magnetic resonance imaging study 

Dear Dr. Gondo:

I'm pleased to inform you that your manuscript has been deemed suitable for publication in PLOS ONE. Congratulations! Your manuscript is now with our production department. 

Kind regards, 

on behalf of

Prof. Kenji Hashimoto 

Section Editor

PLOS ONE